# The Role of Histone Deacetylase 3 Complex in Nuclear Hormone Receptor Action

**DOI:** 10.3390/ijms22179138

**Published:** 2021-08-24

**Authors:** Sumiyasu Ishii

**Affiliations:** Department of Integrative Physiology, Gunma University Graduate School of Medicine, Maebashi 371-8501, Japan; sishii@gunma-u.ac.jp

**Keywords:** histone deacetylase 3, nuclear hormone receptor, nuclear receptor corepressor, silencing mediator for retinoid and thyroid hormone receptors

## Abstract

Nuclear hormone receptors (NRs) regulate transcription of the target genes in a ligand-dependent manner in either a positive or negative direction, depending on the case. Deacetylation of histone tails is associated with transcriptional repression. A nuclear receptor corepressor (N-CoR) and a silencing mediator for retinoid and thyroid hormone receptors (SMRT) are the main corepressors responsible for gene suppression mediated by NRs. Among numerous histone deacetylases (HDACs), HDAC3 is the core component of the N-CoR/SMRT complex, and plays a central role in NR-dependent repression. Here, the roles of HDAC3 in ligand-independent repression, gene repression by orphan NRs, NRs antagonist action, ligand-induced repression, and the activation of a transcriptional coactivator are reviewed. In addition, some perspectives regarding the non-canonical mechanisms of HDAC3 action are discussed.

## 1. Introduction

The endocrine system is one of the major mechanisms that contributes to cell-specific functions and homeostasis. Hormones are produced in the endocrine glands and delivered to the whole body by the bloodstream. Although numerous cells are exposed to these hormones, only target cells that express hormone-specific receptors are able to transduce the signal. Hormones are mainly derived from three origins: peptides, cholesterol, and amino acids. Genes for peptide hormones are encoded in the genome, and the hormones are produced through transcription, translation, and posttranslational modification. These hormones are hydrophilic. Hormones synthesized from cholesterol are lipophilic, and include estrogen, progesterone, testosterone, glucocorticoid, and mineralocorticoid. Hormones derived from amino acids are either hydrophilic or lipophilic. One lipophilic example is thyroid hormone. Despite its lipophilic nature, thyroid hormone is transferred into the cells largely by membrane transporters, such as the monocarboxylate transporter 8 (MCT8) [1], rather than by diffusion. Whereas many peptide hormones bind to membrane-associated receptors, the receptors for a group of small lipophilic hormones mainly regulate the transcription of target genes in the nuclei [2]. These receptors are termed nuclear hormone receptors (NRs). NRs bind to the regulatory region of the target genes, typically as dimers, and regulate transcription in a ligand-dependent manner [2].

NRs constitute a large superfamily with at least 65 members [3,4]. These receptors share a common molecular structure [5,6]. The most highly conserved region is the DNA-binding domain, which contains two zinc finger motifs [7,8]. This region is responsible for the binding of the receptor to the specific DNA sequences, called hormone response elements (HREs), on the target genes. Two zinc finger motifs in NRs typically recognize the hexameric DNA motif PuGGTCA (Pu means A or G), although many exceptions exist. When NRs work as dimers, two hexameric core motifs are located as palindromes, direct repeats, or everted repeats, with or without some spacer nucleotides. The *N*-terminal region is highly variable and harbors an autonomous transcriptional activation function, which is independent of the ligands [9]. The ligand binding domain (LBD) at the C-terminus is responsible not only for ligand binding, but also for receptor dimerization and transcriptional activation or repression of the target genes [10,11]. The transcriptional regulation by NRs is mediated mainly by the interaction with coregulators, including coactivators and corepressors [12]. Basically, NRs suppress transcription in coordination with corepressors, and coactivator interaction is responsible for transcriptional activation by NRs.

Epigenetic histone modifications are highly correlated with transcriptional regulation [13,14]. Among numerous modifications, acetylation of lysine residues in *N*-terminal histone tails is one of the most established marks of an open chromatin structure and transcriptional activation [15,16,17]. Whereas many coactivators exhibit histone acetyltransferase (HAT) activity [18,19], corepressor complexes tend to possess histone deacetylase (HDAC) activity [20,21]. Although there are several HDACs, HDAC3 plays pivotal roles in NR-dependent signal transduction. In addition to tail parts, core histones are also the targets of acetylation [22]. Several types of histone modifications are also involved in transcriptional regulation, such as methylation, phosphorylation, ubiquitylation, glycosylation, and so on [23,24]. These modifications sometimes cooperate with each other and additively or synergistically regulate transcription.

In this review, the basics of the molecular action of NRs will be explained first. The topic will be mainly focused on coregulators and histone acetylation status. Then, the history of HDAC3 as the regulator of NR action will be overviewed. Finally, the molecular mechanisms of NR-HDAC3 functions will be reviewed on each case.

The repressive mechanism of NRs has been mainly explained by corepressor proteins. Although it is established that HDAC3 is the specific core component of the corepressor complex in NR action, the direct involvement of HDAC3 in the NR-dependent repression has not been extensively studied as compared to the roles of corepressors. Given the importance of histone deacetylation by HDAC3, this review will focus on what has been proven about the role of HDAC3 in the NR-dependent repression, rather than the roles of corepressors.

## 2. Basics of Molecular Action of Nuclear Hormone Receptors

### 2.1. Heterodimeric Nuclear Hormone Receptors

NRs bind HREs as heterodimers with a retinoid X receptor (RXR), homodimers, or monomers, and regulate transcription of the target genes [25]. A group of NRs, including a thyroid hormone receptor (TR), a retinoic acid receptor (RAR), and a vitamin D receptor (VDR) form heterodimers with RXR on their target genes. The *N*-terminal regions of these NRs are relatively small. This group of NRs were classically called “type 2 NRs” [26], but this term is no longer commonly used.

In the absence of ligands, these NRs are associated with corepressor complexes that possess HDAC activity and suppress the transcription of the target genes, which is called ligand-independent repression [27,28,29]. Two major corepressor molecules for NRs are a nuclear receptor corepressor (N-CoR) [30] and a silencing mediator for retinoid and thyroid hormone receptors (SMRT, also known as N-CoR2) [31]. These two corepressors are approximately 270 kDa molecules, and share sequence similarities with each other. They also exhibit several common functions, but are not completely redundant, because conventional knockout mice are lethal for both molecules [32,33]. However, these two molecules do not harbor deacetylase activity by themselves. N-CoR and SMRT exhibit HDAC activity as protein complexes that contain HDAC3 (described later). When the ligands are present, corepressors dissociate from NRs, and coactivators are recruited instead (Figure 1a) [34,35]. These coactivators with HAT activity include the p160 family [36,37,38,39], which further recruit other HATs, such as the CREB binding protein and p300 [40,41]. This exchange is mainly due to the conformational change of NRs induced by ligands. The unliganded NRs favor binding to corepressors. Upon ligand binding, the LBD undergoes a structural transition called the “mousetrap” mechanism, which allows the interaction with coactivators [42]. The LBDs of NRs consist of 12 helices. Helix H12 is the extreme C-terminal part of NRs, and extends toward the outside without a ligand. Upon the binding of ligands, helix H12 changes its position and covers the core LBD region like a “mousetrap”. The cofactor exchange does not actually follow such an “all-or-none” switch model. It was revealed in vivo that the expression of TR-target genes is regulated by a shift in the relative binding of corepressors and coactivators [43]. It should be noted that these NRs similarly work as homodimers, depending on the target genes.

### 2.2. Homodimeric Nuclear Hormone Receptors

The typical homodimerization NRs include the estrogen receptor (ER), the progesterone receptor (PR), the androgen receptor (AR), the glucocorticoid receptor (GR), and the mineralocorticoid receptor (MR). These NRs have relatively large *N*-terminal domains. They were previously called “type 1 NRs” [26], which is no longer a common term. In the absence of ligands, they bind to chaperone molecules, such as heat shock protein (HSP) 90 and HSP 70 in the cytosol [26,44]. In addition, unliganded ER is also localized to the nucleus, but does not bind its target genes [45]. Therefore, these NREs do not directly regulate gene transcription in the unliganded state. Upon binding to the ligands, NRs bind DNA, and stimulate transcription in association with coactivators (Figure 1b). The conformational change by the “mousetrap” mechanism [42] is responsible for the interaction with coactivators. In the traditional model, it was believed that HSPs dissociate from NRs and remain in the cytosol. However, recently it was shown that HSPs play an important role in the nuclear translocation of NRs [45,46]. Moreover, HSPs in the nuclei contribute to the transcriptional activity of liganded NRs [47]. In this model, the role of corepressors with HDAC activity is limited.

### 2.3. Orphan Nuclear Hormone Receptors

In addition to these NRs, there are multiple orphan NRs whose ligands are yet to be identified [48]. These NRs serve as transcription factors and regulate gene transcription. Whereas some of them activate transcription of the target genes in cooperation with coactivators, others serve as transcriptional repressors by interacting with corepressors. Orphan NRs are attractive therapeutic targets of diseases because of their important roles and high specificities for their target genes. Many efforts are underway to identify natural or synthetic ligands for several orphan NRs.

## 3. Histone Deacetylase 3 in Nuclear Hormone Receptor Corepressor Complex

### 3.1. Basics of Histone Deacetylase 3

Although acetylation of histones has been extensively studied, histones are not the only substrates for acetylation and deacetylation. Given the importance of acetylation of non-histone proteins, HDACs have been renamed to lysine deacetylases (KDACs) [49,50]. Similarly, HATs are officially called lysine acetyltransferases (KATs). However, the familiar name “HDAC” will be used in this review.

Since the discovery of HDAC1 [20], at least 18 potential HDACs have been reported in humans. Based on their sequence similarity, human HDACs are categorized into four classes [51]. Class I HDACs are similar to yeast Rpd3, and include HDAC1, 2, 3, and 8. The members of class II HDACs include HDAC4, 5, 6, 7, 9, and 10, and are homologous to yeast Hda1. Unlike other classes, the activity of class III HDACs is dependent on nicotinamide adenine dinucleotide (NAD^+^). Class III HDACs show similarities to yeast Sir2, and include SIRT1, 2, 3, 4, 5, 6, and 7. HDAC11 is categorized as class IV because the sequence in the catalytic domain is unique.

HDAC3 was initially reported soon after HDAC1 [52,53,54]. HDAC3 consists of 428 amino acids, and the theoretical molecular weight is 49 kDa. It is ubiquitously expressed in many cell lines and tissues. HDAC3 does not directly bind to DNA or NRs. Bhaskara et al. reported that HDAC3-knockout mice were embryonic lethal, and that HDAC3-deficient embryonic fibroblasts exhibited a delay in cell cycle progression, cell cycle-dependent DNA damage, and apoptosis [55]. Indeed, HDAC3 plays pivotal roles in maintaining basic properties of cells, including Aurora B kinase activity in mitosis [56]; mitotic kinetochore–microtubule attachment [57]; sister chromatid cohesion [58]; maintenance of chromatin structure and genome stability [59]; DNA replication in hematopoietic progenitor cells [60]; and gap 2/mitosis progression [61].

### 3.2. Histone Deacetylase 3 and Nuclear Hormone Receptor Corepressors

HDAC3 was implicated in nuclear hormone receptor action by the identification of its core interaction partners in the early 21st century. Biochemical purification of HDAC3-associated protein complex revealed that N-CoR and SMRT are the members of the HDAC3 core complex [62,63]. Reciprocally, N-CoR and SMRT form a stable complex with HDAC3, but not with other HDACs [63,64,65,66]. These findings were surprising because many researchers expected that well-known HDAC1 and 2 would play a major role; however, these findings were confirmed because the same results were reported by several independent laboratories. Currently, it is established that HDAC3 core complex is distinct from HDAC1/2 core complex, which contains RbAp46 and RbAp48. These facts indicate that HDAC3 is the main HDAC responsible for the ligand-independent repression by NRs, because both N-CoR and SMRT are the major mediators of repression, but do not exhibit deacetylase activity by themselves. On the other hand, HDAC3 does not directly bind NRs. N-CoR and SMRT tether HDAC3 to transcriptional repressors, like unliganded NRs.

Moreover, interaction with N-CoR/SMRT substantially potentiates the enzymatic activity of HDAC3. The deacetylase activating domain (DAD) is determined for both N-CoR and SMRT [67]. The DADs in the corepressors are also responsible for the interaction with HDAC3, and contain SANT motifs, which contribute to histone binding in other proteins. Analyses of the crystal structure of DAD-HDAC3 complex revealed that inositol tetraphosphate is necessary for the interaction [68]. An introduction of mutations that abolish HDAC3 interaction in both N-CoR and SMRT results in a loss of deacetylase activity in vivo [69]. Indeed, integration of HDAC3 into N-CoR/SMRT complex is important for the ligand-independent repression by NRs [70]. One of the additional factors that regulate enzymatic activity is phosphorylation of HDAC3. Casein kinase 2 and protein serine/threonine phosphatase 4 complex are involved in this regulation [71].

Other HDAC3 core complex members include transducin beta-like 1 X-linked (TBL1X), TBL1X receptor 1 (TBL1XR1), and G protein pathway suppressor 2 (GPS2). These molecules also play important roles in the functions of the corepressor complex [72]. Both TBL1X and TBL1XR1 are WD40 repeat-containing proteins. Suggested roles of these molecules in N-CoR/SMRT complex include histone binding [73] and degradation of the corepressor complex via the ubiquitin–proteasome pathway [74]. Mutations in the *TBL1X* gene are associated with central hypothyroidism [75], although the mechanism is not clarified yet. GPS2 is responsible for the inhibition of the JNK pathway by HDAC3 complex [63]. On the other hand, it was reported that phosphorylation of c-Jun, an AP-1 transcription factor subunit, by JNK pathway dissociates HDAC3 complex from c-Jun and relieves transcriptional repression [76].

### 3.3. Physiological Roles of Histone Deacetylase 3

Tissue-specific functions of HDAC3 have been studied using cell-type specific knockout mice [77], because conventional knockout mice are embryonic lethal [55,78]. For example, HDAC3 is involved in energy metabolism and organogenesis of the heart [78,79,80]. Liver-specific knockout mice revealed the roles of HDAC3 in lipid metabolism and circadian rhythm [81,82,83]. HDAC3 is also involved in the alternative activation of the macrophages [84]. HDAC3 in intestinal epithelial cells contributes to local lymphocyte activation [85]. The roles in bone formation [86,87], brain organogenesis [88], and lung development [89] are documented. Furthermore, HDAC3 regulates thermogenic gene transcription in brown adipose tissue [90], insulin sensitivity in muscles [91], and insulin secretion [92]. However, there is no report of human diseases caused by a germline mutation of *HDAC3* gene so far. This is probably because these mutations result in embryonic lethality.

In addition, it is suggested that HDAC3 is involved in cancer progression. Overexpression of HDAC3 is observed in many types of cancers, particularly leukemia and lymphoma [93]. Moreover, several broad-spectrum HDAC inhibitors have been approved by the FDA as therapeutic agents for the treatment of particular types of lymphomas [94]. Therefore, specific inhibition of HDAC3 is an attractive therapeutic strategy for the treatment of malignant diseases [93].

## 4. Histone Deacetylase 3 in Nuclear Hormone Receptor Action

### 4.1. Ligand-Independent Repression

As mentioned above, a heterodimeric group of NRs, including TR, RAR, and VDR, is associated with N-CoR/SMRT complexes that contain HDAC3 and suppress the transcription of the target genes in the absence of a ligand. On the other hand, the homodimerization NRs, including ER, PR, GR, MR, and AR, are “neutral” in terms of gene expression without ligands, because these NRs do not bind their target genes in the unliganded state. It seems that the importance of the difference between these two groups is not as highly evaluated as before. However, it is a notable point when considering the function of N-CoR/SMRT and HDAC3.

The findings show the importance of ligand-independent repression by the heterodimeric NRs, including its role in phenotypic expression of TR-knockout mice and hypothyroid mice. The defects are much more severe in hypothyroid mice than in TR-deficient mice, indicating the roles of repression by unliganded TR [95,96]. As a core deacetylase component of N-CoR/SMRT complex, HDAC3 plays a critical role in ligand-independent repression by deacetylating histone tails (Figure 2a). Indeed, it has been reported that HDAC3 is involved in the repressive functions of unliganded TR [65,97,98], RAR [99], and VDR [100]. In addition to these reports, the role of N-CoR/SMRT in the ligand-independent repression are extensively studied in numerous cases. Although the involvement of HDAC3 is not directly examined, it is expected that HDAC3 is the major player in these cases.

RARα is involved in the etiology of acute promyelocytic leukemia (APL), in which leukocyte differentiation is blocked at the promyelocyte stage. Retinoic acid plays important roles in the differentiation of many types of cells, including granulocytes. The majority of APL cases are caused by chromosomal translocations that produce a promyelocytic leukemia (PML)-RAR alpha (RARA) fusion protein [101]. PML-RARA disrupts both PML nuclear body assembly and RAR-dependent transcription [102]. The fusion protein competes with wild-type RAR in a dominant-negative manner and suppresses the transcription of retinoic acid-dependent differentiation genes. The PML-RARA protein is associated with N-CoR/SMRT complex like unliganded RAR, and HDAC3 is involved in the gene repression as the main HDAC [103,104,105]. Treatment with supraphysiological doses of all-trans retinoic acid dissociates N-CoR/SMRT-HDAC3 complex from the fusion protein and induces gene expression and differentiation of the tumor cells. Retinoic acid also enhances the formation of PML nuclear body assembly, which is also potentiated by treatment with arsenic trioxide.

### 4.2. Gene Repression by Orphan Nuclear Hormone Receptors

Gene expression is suppressed by several orphan NRs, whose ligands have not yet been identified. These NRs work as monomers, homodimers, or heterodimers with RXR or other NRs [48]. The typical molecular mechanism is similar to ligand-independent repression, where histone deacetylation by N-CoR/SMRT-HDAC3 complex plays a pivotal role.

An orphan NR, REV-ERBα, was initially identified as a molecule whose gene is encoded by the reverse strand of the *c-erbAα* oncogene [106]. Later, it appeared that the *c-erbAα* oncogene encodes TRα [107,108]. REV-ERBα works as a transcriptional repressor in coordination with the N-CoR complex. REV-ERBα plays a central role in circadian rhythm by repressing core clock genes, including BMAL1. The critical involvement of HDAC3 was reported on a genome-wide scale [109].

Regarding other orphan NRs, HDAC3 is also involved in lineage restriction of embryonic stem cells mediated by dosage-sensitive sex reversal, adrenal hypoplasia critical region, chromosome X, gene 1 (DAX1) [110], TR2-mediated gene repression [111], and neural stem cell proliferation by TLX [112]. Although direct evidence is still missing, other repressive orphan NRs are speculated to cooperate with HDAC3, because some of them are associated with N-CoR/SMRT [113].

### 4.3. Nuclear Hormone Receptor Antagonist Action

As shown in Figure 1b, a homodimeric group of NRs is not associated with the target genes in the absence of a ligand. The HDAC3 complex is not directly involved in that model; however, some antagonists for these NRs induce gene repression in coordination with the corepressor complex, including HDAC3 (Figure 2b).

Some of the most extensively studied NR antagonists are the modulators of ER function used in the treatment of hormone-sensitive breast cancer [114]. Selective estrogen receptor modulators, such as tamoxifen and raloxifene, show antagonistic effects, at least to some extent [115]. HDAC3 in the N-CoR/SMRT complex is involved in tamoxifen-mediated gene repression [116]. In contrast, the role of N-CoR/SMRT-HDAC3 complex in the treatment of hormone-sensitive prostate cancer with selective androgen receptor modulators is not supported so far [117].

### 4.4. Ligand-Induced Repression

Typical NR agonists stimulate the transcription of the target genes, as mentioned above. However, a certain number of genes are suppressed in the presence of hormones. It is hard to explain the molecular mechanism behind this because liganded NRs bind coactivators rather than corepressors due to their structures [42]. Although some reports suggest the role of HDAC3 in ligand-induced repression [118,119], a common mechanism that explains all ligand-dependent repression cases is yet to be described.

### 4.5. Activation of Transcriptional Coactivator by Histone Deacetylase 3

Although histone deacetylation is highly associated with gene repression, there are several genes activated by HDAC3 via NRs. At least some cases are explained by deacetylation of non-histone proteins [50]. One convincing example is the role of HDAC3 in transcriptional stimulation in brown adipose tissue [90]. In this model, the N-CoR/SMRT-HDAC3 complex is tethered to chromatin by the first identified orphan nuclear hormone receptor, the estrogen-related receptor-α (ERRα) [120]. ERRα plays an essential role in adaptive thermogenesis by enhancing the transcription of thermogenic genes [121]. Peroxisome proliferator-activated receptor-γ coactivator 1α (PGC1α) is an essential coactivator for ERRα [122], and the function of PGC1α is inhibited by acetylation [123]. A class III HDACs SIRT1 is a known HDAC that deacetylates and activates PGC1α [124]. In addition, Emmett et al. reported that HDAC3 stimulates thermogenic gene expression by deacetylating PGC1α (Figure 2c) [90].

## 5. Non-Canonical Mechanisms of Histone Deacetylase 3 Action for Other Transcription Factors

Although the responsible transcription factors are unknown or are not NRs, non-canonical mechanisms of HDAC3 action are reported. These mechanisms might be involved in NR action as well.

For example, several functions that are independent of the enzymatic activity of HDAC3 were studied. HDAC3 tethers target genes to the peripheral region of nuclei by interacting with the nuclear lamina protein lamina-associated polypeptide 2 [80,125]. HDAC3 recruits polycomb repressive complex 2, which possesses histone methyltransferase activity, independently of HDAC activity [126]. In addition, HDAC3 is recruited to activate transcription factor 2-bound sites without association with N-CoR/SMRT, and activates the expression of inflammatory genes [127].

Many findings of acetylation/deacetylation of non-histone proteins are emerging as novel and important aspects of posttranslational modifications [49,50]. Several non-histone substrates are reported for HDAC3. For instance, p65, a subunit of nuclear factor kappa-B (NF-κB), is deacetylated by HDAC3. Deacetylation results in stimulation or suppression of NF-κB-dependent transcription, depending on the acetylation sites [128,129]. Sex-determining region Y (SRY), a key transcription factor for the determination of testis differentiation, also undergoes deacetylation by HDAC3 and subsequent nuclear export [130].

In addition, HDAC3 does not localize only in nuclei. It is also identified in the cytoplasm of several cell lines, including DT40 [131], 293 [132], mouse embryonic fibroblasts [133], 3T3-L1 [133], and PC3 [134]. The cytosolic function of HDAC3 is still under investigation.

## 6. Conclusions and Perspectives

As the core HDAC component of the N-CoR/SMRT corepressor complex, HDAC3 is largely responsible for transcriptional repression mediated by NRs and other transcription factors. The HDAC3 complex mainly suppresses transcription by deacetylating histones.

On the other hand, recently the roles of HDAC3, independent of its enzymatic activity, are reported for transcription factors other than NRs, as mentioned above. In addition, non-histone proteins are of interest as novel substrates for HDAC3. These tendencies suggest that non-canonical mechanisms of HDAC3 in NR action will be a hot topic in the near future. Moreover, HDAC3 is also localized in the cytosol. Therefore, it would not be surprising if cytosolic NR-HDAC3 complex exerts some transcription-independent functions.

## Figures and Tables

**Figure 1 ijms-22-09138-f001:**
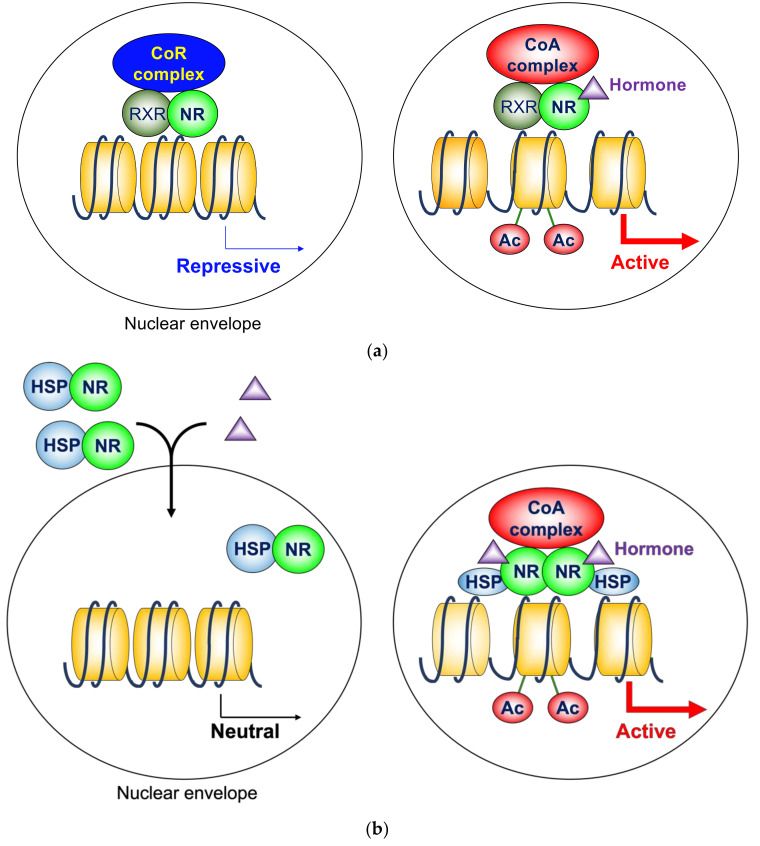
(**a**) Cofactor exchange by heterodimeric nuclear hormone receptors. In the absence of ligands, these NRs are associated with corepressor complexes that possess HDAC activity and suppress the transcription of the target genes, which is called ligand-independent repression. When the ligands are present, corepressors dissociate, and coactivators are recruited instead. Many of these coactivators possess HAT activity. (**b**) Molecular action of homodimeric nuclear hormone receptors. In the absence of ligands, they bind to chaperone molecules, such as HSP 90 and HSP 70 in the cytosol. In addition, unliganded ER is also localized to the nucleus, but does not bind its target genes. Upon binding to the ligands, NRs enter nuclei, bind DNA, and stimulate transcription in association with coactivators. HSPs play important roles in nuclear translocation of NRs and transcriptional activity of liganded NRs. CoR, corepressor; CoA, coactivator; NR, nuclear hormone receptor; RXR, retinoid X receptor; Ac, acetylation; HSP, heat shock protein.

**Figure 2 ijms-22-09138-f002:**
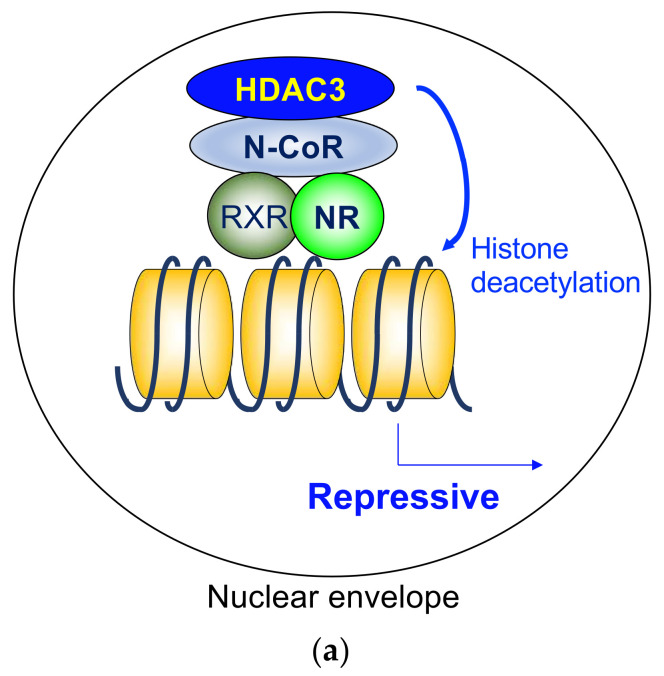
(**a**) Histone deacetylase 3 in ligand-independent repression. As a core deacetylase component of N-CoR/SMRT complex, HDAC3 plays a critical role in ligand-independent repression by deacetylating histone tails. (**b**) Histone deacetylase 3 in antagonist-induced repression. Some antagonists for homodimeric NRs induce gene repression in coordination with the HDAC3 complex. (**c**) Histone deacetylase 3 activates a transcriptional coactivator by deacetylation. The activated coactivators stimulate transcription on the target genes. HDAC3, histone deacetylase 3; N-CoR, nuclear receptor corepressor; NR, nuclear hormone receptor; RXR, retinoid X receptor; HSP, heat shock protein; CoA, coactivator.

## Data Availability

No new data were created or analyzed in this study. Data sharing is not applicable to this article.

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
