# Peer review of "The Role of Histone Deacetylase 3 Complex in Nuclear Hormone Receptor Action"

_ijms, 2021, doi:10.3390/ijms22179138_

Round 1

Reviewer 1 Report

This review is comprehensive, well-written, and an important addition to the existing literature on HDAC3 and nuclear hormone receptors. The review would be further strengthened by addressing the following points:

Line 30: Add "and" to the list: "glucocorticoid, and mineralocorticoid"

Line 31: Despite being lipophilic, thyroid hormone does not diffuse through the plasma membrane.  For accuracy, it should be noted that the transmembrane passage of thyroid hormone in cells is mediated by specific thyroid hormone transporters, such as MCT8.

Line 34: Although, "nuclear receptors" is usually the preferred term in the literature, the author should be consistent throughout the review. Since "nuclear hormone receptor" is used primarily in this review, here NR should be defined as "nuclear hormone receptor." Note that the figure legends will also need to be corrected ("NR: nuclear receptor" should be replaced with "NR: nuclear hormone receptor").

Line 53: Specify "N-terminal histone tails"

Line 62: Cite some references for the preceding information.

Line 68: Make plural: "nuclear hormone receptors"

Line 72: usually RAR is called the "retinoic acid receptor"

Line 111-115: This is an oversimplification and should be updated for accuracy. For example, the estrogen receptor is primarily localized to the nucleus in both the absence and presence of ligand, but only associates with DNA when liganded. Also, there is now evidence that HSPs play a role in nuclear translocation of GR/MR/AR and complexes exist in the nucleus. The caption for Figure 2 should also be updated.

Line 221-223: The statement that "On the other hand, the unliganded homodimerization NRs, including ER, PR, GR, MR, and AR, are “neutral” in terms of gene expression because they are outside of nuclei" is also an oversimplification, in particular for ER and PR, since they can both be primarily nuclear in the absence of ligand (although not interacting with target genes).

Line 341-342: It would be more informative to expand this sentence to indicate in what cell/tissue types HDAC3 has been shown to be cytosolic.

Line 355: Add "the" so that sentence reads: "are studied further in the future"

Reviewer 2 Report

Review: ijms-1307775

In this work, the authors have reported the role of histone deacetylase 3 complex in nuclear hormone 2 receptor action. This manuscript is important for the study of nuclear hormone 2 receptor. However, publication of this manuscript in its present form is not recommended. It contains a number of confusing statements. To be considered further for publication, this work will need to be more organized and condensed in support of the claims made in the paper. Some specific points of concern are noted below:

1) The MS needs to be condensed and mature. A schematic/cartoon summarizing all the pathways is preferred.

2) The uniqueness of this model needs to be explained with more clarity. Briefly explain how this concept is unique compare to previously published reports

3) The last sentence of the conclusion should be reworded.

Minor comments:

The ligand binding domain should be LBD on pg.2 line 90.

Round 2

Reviewer 2 Report

The author has addressed the issues from the first review. The manuscript is ready for publication without any further revision.